# Study of real-time parameter measurement of ring rolling pieces based on machine vision

**Xiaoge Fu**[ID], **Han Li, Zhijiang Zuo\*, Libo Pan**

State Key Laboratory of Precision Blasting, Jianghan University, Wuhan, 430056, China

\* zzjdfcy@163.com

## Abstract

Real time parameter measurement cannot be carried out to dynamic ring parts during automation ring rolling processes so that rolling process parameters cannot be adjusted in time. Considering effects of shaping of ring rolling parts, a visual measurement platform was set up and a machine vision-based non-contact real -time measurement method was put forward. This article improves the subpixel level edge extraction algorithm to extract edge data information of circular rolling pieces. Based on the characteristics of circular rolling pieces, an RG-Hough transform method is proposed to fit the detected edge data information. The conversion relationship between pixel and actual sizes were determined in combination with the camera calibration to gain parameters of ring rolling parts. Measurements of ring parts (OD: 462.12mm; and ID: 315.53mm) were applied to verify the effectiveness of our method. Our measurement error is ±0.25mm and our average speed can be up to 104ms/frame. Our study can provide powerful technical supports for intelligent control of ring rolling pieces.

**Data Availability Statement:** All relevant data are within the manuscript and its Supporting Information files.

## 1 Introduction

Ring parts are very popular in mechanical devices such as bearing rings, aerospace cases, rockets and fans in petrochemicals, aerospace and aviation [1], ships, energy, vehicles and other industrial fields [2,3], which are usually required against impact, high temperature, corrosion and so on and whose quality also directly affects the overall level and reliability of major equipment. Thus, they are regarded as the prerequisite for development of advanced systems and one important factor of industry development and advanced levels for any country [4]. Rolling of ring parts is a continuous partially loaded molding of blank ring parts [5,6].

The rolling principle of ring parts is shown in Fig 1 [7], where a driving roller rotates continuously, a core roller feeds in a radial straight line and a cone roller rotates continuously for axial feeding so that the radial wall thickness and axial height of a ring part can fall while its diameter can continuously grow; thus, a continuous local plastic deformation can occur in its cross-section contour [8,9]. Measurement is performed during its rolling process to determine whether its design sizes can be met and control of its rolling process is necessarily required in the later rolling stage. The rolling equipment can be adjusted in time.

There are two main existing methods of measuring the dimensions of rolled rings [10]. As for the former, contact measurement of pieces are primarily carried out and it is applicable to small batches of production and those cases with low requirements for measurement speed.

**Funding:** Jianghan University Science and Technology Innovation Project (No.2021kjzx011); Jianghan University Graduate Research and Innovation Fund Project (No.KYCXJJ202306).

**Competing interests:** The authors have declared that no competing interests exist.

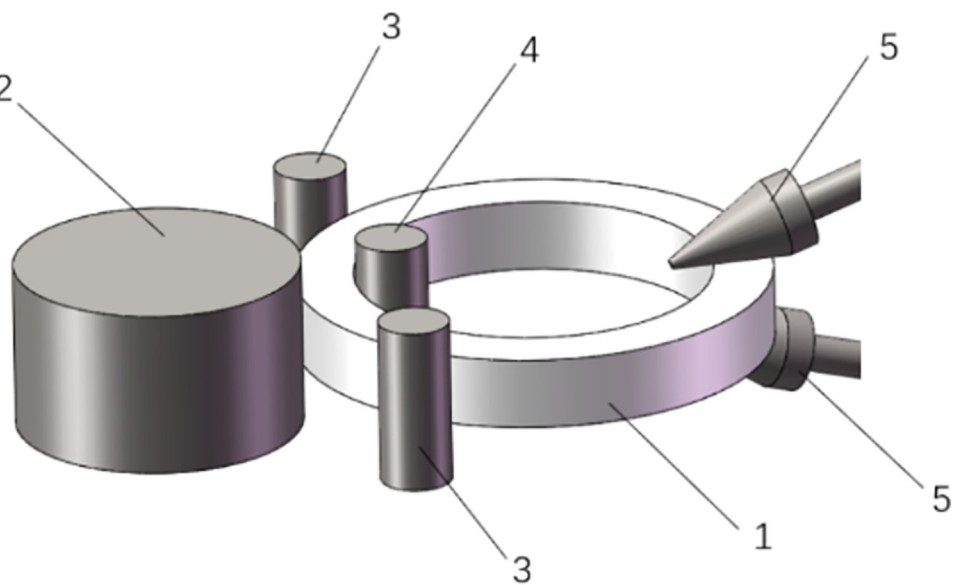

**Fig 1. Basic rolling principle of ring parts.** 1—Ring part; 2—Driving roller; 3—Guiding roller; 4—Core stick; and 5—Cone roller.

Whereas, measurements are limited for ring rolling parts under high temperature so that its measurement efficiency can be relatively low and results may be in relatively poor accuracy. The measurement surface may be easily damaged and tools are also worn. Also, testing workers may be harmed to a certain extent. On the other hand, the latter primarily includes those conventional methods such as grating, electromagnetic, sound wave and laser measurements [11,12]. Zheng Lu et al. [13] obtained point cloud data by laser scanning, processed the point cloud data by multiple regression filtering algorithm, and measured the outer diameter of the ring. Shimizu S et al. [14] applied the industrial CT method to the high-precision measurement of elliptic shape. This method reconstructed the three-dimensional morphology of the elliptic mirror through tomography, and then fitted its shape parameters through advanced algorithms. Yucun Zhang et al. [15] scanned multiple angles by laser scanner, obtained complete radial cross-section data of the ring through stitching, and converted the data into a complete three-dimensional ring model. However, these methods are prone to interference from the temperature of the ring itself and the ambient temperature, resulting in loss of measurement data, thus affecting the accuracy of ring size measurement [16].

Machine vision-based measurement is a new type non-contact method, not only whose accuracy and efficiency are high but also whose stability is improved [17,18]. Moreover, parts cannot be damaged. Thus, it is becoming more popular in the measurement field. Bangguo Wang et al. [19] used two monochrome charge-coupled device (CCD) cameras to acquire images of hot forgings, and realized length measurement of hot forgings through binocular vision system. Zhang Xuelin [20] utilized a industrial camera to acquire images and completed measurements after a series of mathematical morphology processing in this image processing module so as to get measurements. Xianbin Fu et al. [21] established 3D point cloud data and 3D model of ring forgings by using point cloud images and 2D laser scanner, and derived the size of ring parts. Yazid Saif et al. [22] used image processing techniques to analyze the surface of the model through circular dimensional analysis, which can detect the quality problems of the ring. In this respect, the machine vision-based measurement technique is very effective and it can be put into operation in a harsh environment.

In response to above problems, a real-time parameter method is put forward based on machine vision for measurement of ring rolling parts and a visual measurement platform is established. Considering characteristics of ring rolling parts, improved the traditional Zernike moment subpixel edge detection according to the characteristics of ring rolled parts, and proposed the RG-Hough transform method, and parameters of ring rolling parts are measured in combination with our camera calibration results. Our study can be regarded as preliminary work for further intelligent control of ring parts rolling.

The main contributions of this paper are summarized as follows:

1. A visual measuring platform for hot rolled ring rolling was designed. Based on the characteristics of the rolling process, the platform selects the appropriate type of industrial camera, lens, light source, and motor, and designs the appropriate stage, rotating platform and rolling motion structure, which can produce the ring rolling conditions.

2. A non-contact real-time measurement method based on machine vision was proposed in view of the harsh rolling environment and the peeling of oxide during rolling. This method includes improving the traditional Zernike moment subpixel edge detection, combining with the characteristics of circular rolled parts, RG-Hough transform is proposed, which provides better measurement effect and faster detection speed for the overall measurement of circular rolled parts.

3. Developed the upper computer software for real-time parameter measurement of ring rolling process. The software integrates the visual measurement platform control system and the visual measurement system, through which the software can control the operation of the visual measurement platform, measure the ring rolled parts in real time, display the original drawing of the ring rolled parts, the visual algorithm result diagram, and the real-time measurement value, and display the measured results in real time.

## 2 Conceptual designs

### 2.1 Overall scheme

The purpose of our measurement system indicates that our measurements shall mainly focuses on the inner and outer sizes of ring parts and they shall be real-time and accurate. The overall scheme of the measurement system is mainly divided into four steps, as shown in Fig 2.

Firstly, an appropriate industrial camera, lens, light sources, and motor models are necessarily selected and a proper objective table, rotating platform and simulation rolling structures are designed for our visual measurement in combination with the ring rolling process. Secondly, the camera internal parameters and position parameter model shall be calculated out based on experiments to get the relationship between the pixel and world coordinate systems

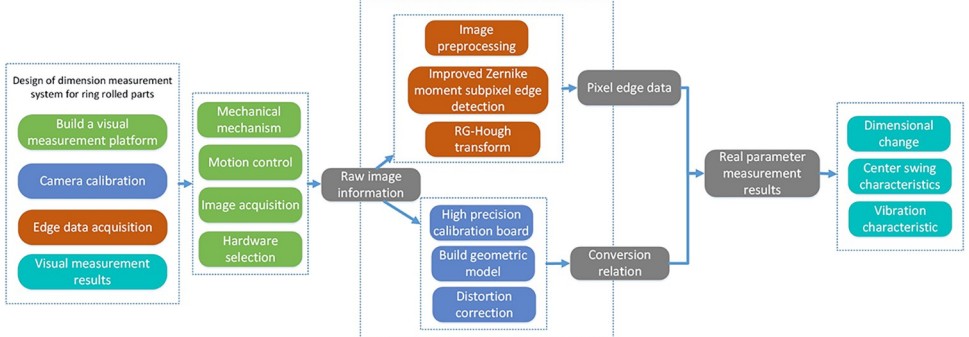

**Fig 2. Overall scheme of measurement system.**

of ring parts. The position of the central point of the calibration board is substituted into the conversion relation in the image coordinate system to calculate out its corresponding theoretical coordinates in the world coordinate system. Thirdly, images of the high-speed ring part are acquired by means of the image collection module and transmitted to the computer. The relevant image pre-processing operation of the collected images is performed to highlight the contour structure of its effective area. The sub-pixel edge coordinate data of more accurate rolling parts can be subsequently gained by means of our improved Equation pixel edge detection algorithm. Then, edges of a ring rolling part cam be fitted by means of the Hough transformation and based on its geometric characteristics. Its center is positioned and analysis is performed to the corresponding rolling center data to gain the real parameter measurements during the entire rolling process. Lastly, through the development of the upper computer software to display the measured results visually, more intuitive observation of the rolling process of the ring rolling parameters change.

## 2.2 Mechanical structure design

The mechanical structure mainly includes a vision module, a rotating module, a positioning runout module, and an impact module, as shown in Fig 3. The main function of the vision module is to carry out a series of operations through the camera to obtain the data that

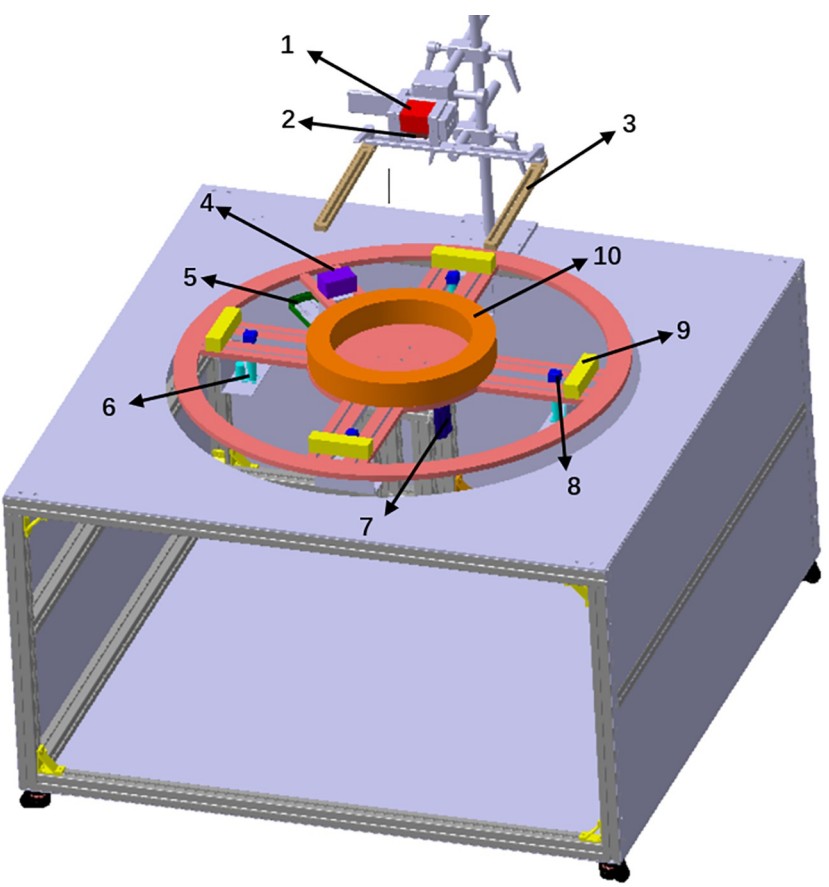

**Fig 3. Mechanical structure design schematic diagram.** 1—Industrial camera; 2—Lens; 3—Light sources; 4—Collision block; 5—Linear motors; 6—Motorized expansion link; 7—Rotating platform; 8—Rubbing block; 9—Locate the block; and 10—Ring part.

characterizes the dimensional change characteristics of the ring and complete the real-time measurement of the dynamic dimensional characteristics of the ring. The main function of the rotating module is to drive the ring and other parts to rotate, so that the ring can reach a stable rotating state. The main function of the positioning runout module is to tighten the ring through the positioning block, and use the electric telescopic rod to bump the ring during the rotation process. The main function of the impact module is that when the ring is rotating, the motor drives the impact block to impact the ring, so that the ring is offset. The ring is placed on the working plane, the working plane relates to the rotating platform, and the rolling process of the ring is simulated by rotating the module. The impact module and the positioning runout module impact the ring in the horizontal and vertical directions, and the scene of automatic deviation and runout of the ring in the processing is simulated. The lens relates to the industrial camera and placed above the ring to collect the ring. The computer uses the vision module to measure the image collected by the data acquisition card in real time.

Four types of working conditions can be simulated in our vision-based ring rolling part platform, which are size measurement and real-time monitoring of growth sizes of under a stable rotation, runout tracing and offset correction during a rotation process.

1. Dimension measurement of ring rolled parts under stable rotation

Firstly, the ring-shaped rolled parts are placed on the ring inspection bearing platform, and the ring-shaped rolled parts are fixed by the positioning block, and the ring-shaped rolled parts are rotated by the rotary module. The rotary module can be adjusted to different rotational speeds, and at the same time, the vision module is used to measure the dimensions of the ring-shaped rolled parts under different rotational speeds.

2. real-time monitoring of the growth size of the ring rolled parts under stable rotating conditions

In the process of production of ring rolled parts, its diameter will continue to grow, through the linear mobile module to drive the rotary module left and right, the vision module on the ring rolled parts part of the region detection, to get the current state of the ring rolled parts and size information.

3. the rotation process of the ring-shaped rolled parts of the jump to follow

Ring rolled parts in the growth process, there are some areas due to uneven force resulting in the ring rolled parts of the temporary jumping situation. Jumping principle of operation for when the ring-shaped rolled pieces in the stable rotation, the use of electric telescopic rod installed on the top block of the ring-shaped rolled pieces for the top impact.

4. the rotation process of the ring-shaped rolling pieces offset correction

Ring rolled parts in the rolling process, as its rolling process is the use of extrusion ring rolled parts lead to its thickness becomes thin diameter becomes large. Therefore, in the process of extrusion, the ring rolled piece exists in the left and right deviation of the situation. When the ring rolled piece of stable rotation, the impact module on the ring rolled piece of impact, change the position of the ring rolled piece. When the position of the ring-shaped rolling member is changed, the vision module monitors the position change of the ring-shaped rolling member, and adjusts the position of the ring-shaped rolling member through the linear movement module, to readjust its position to a suitable position in the field of view of the camera of the vision module.

## 2.3 Software design

The software measurement part designed in this paper mainly includes seven parts: image acquisition, camera calibration, image preprocessing, edge detection of ring pieces, edge fitting of ring pieces, error compensation, and result display and saving. The software flow is shown

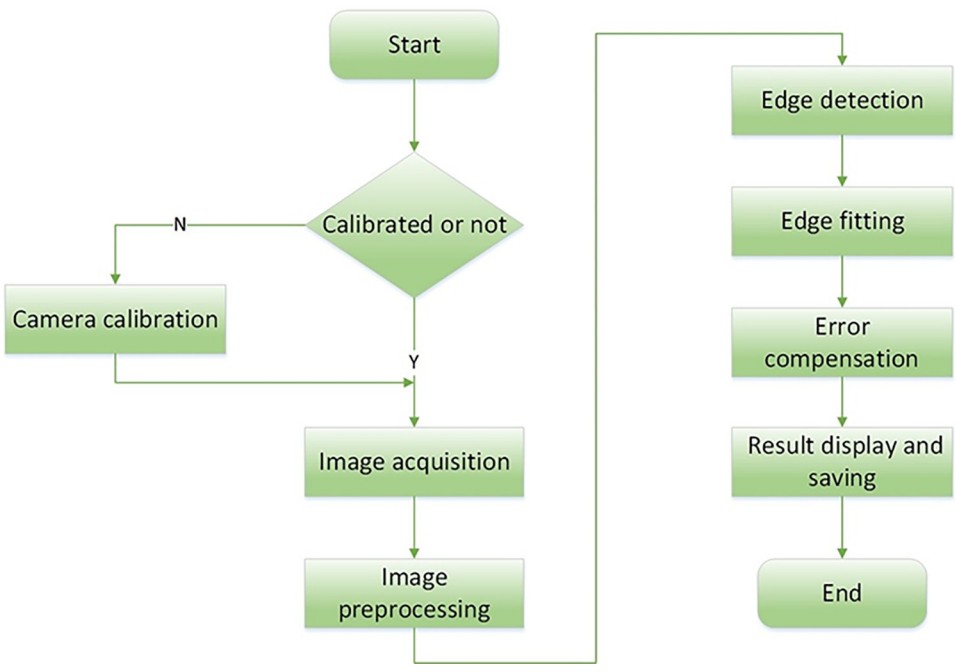

**Fig 4. Software process.**

in Fig 4. Through the automatic focusing of the camera to the ring piece, the image acquisition of the ring piece in the process is carried out through the software, and the distortion image is corrected by using the camera calibration method to get the distortion-free ring piece image. A series of preprocessing operations are performed on the ring piece image to eliminate the interference of noise and obtain a higher quality ring piece image. Then pixel-level edge detection and sub-pixel edge detection are performed on the ring pieces in the image to eliminate the occlusion of the core roller and the drive roller, to obtain the required edge information of the ring pieces, and to fit the incomplete edge information of the ring pieces to obtain the edge information of the complete ring pieces, and to obtain the measured data of the ring pieces. According to the thermal deformation law formula and the error formula, the measured data are compensated for the error, and the final ring parameters are obtained. Finally, the results are displayed in real time on the user interactive interface, and the results are saved in real time in Excel table according to the chronological order, which is convenient for analysis and comparison at a later stage.

## 3 Measurement algorithm

### 3.1 Image preprocessing

As the ring is growing and changing during the rolling process, accompanied by swinging and jumping, and the oxidized skin on the surface of the ring will come off during the process, it will lead to the ring image acquired by the vision measurement system to contain noise, cause the true edge of the ring cannot be extracted. The result of extracting the edge of the image directly is shown in Fig 5.

To improve the quality of the captured ring image and extract the edge data of the ring more accurately, it is necessary to eliminate the interference of noise, so it is necessary to preprocess the captured ring image to reduce the interference of irrelevant information in the image and enhance the edge information [23]. Noise will greatly affect the result of visual

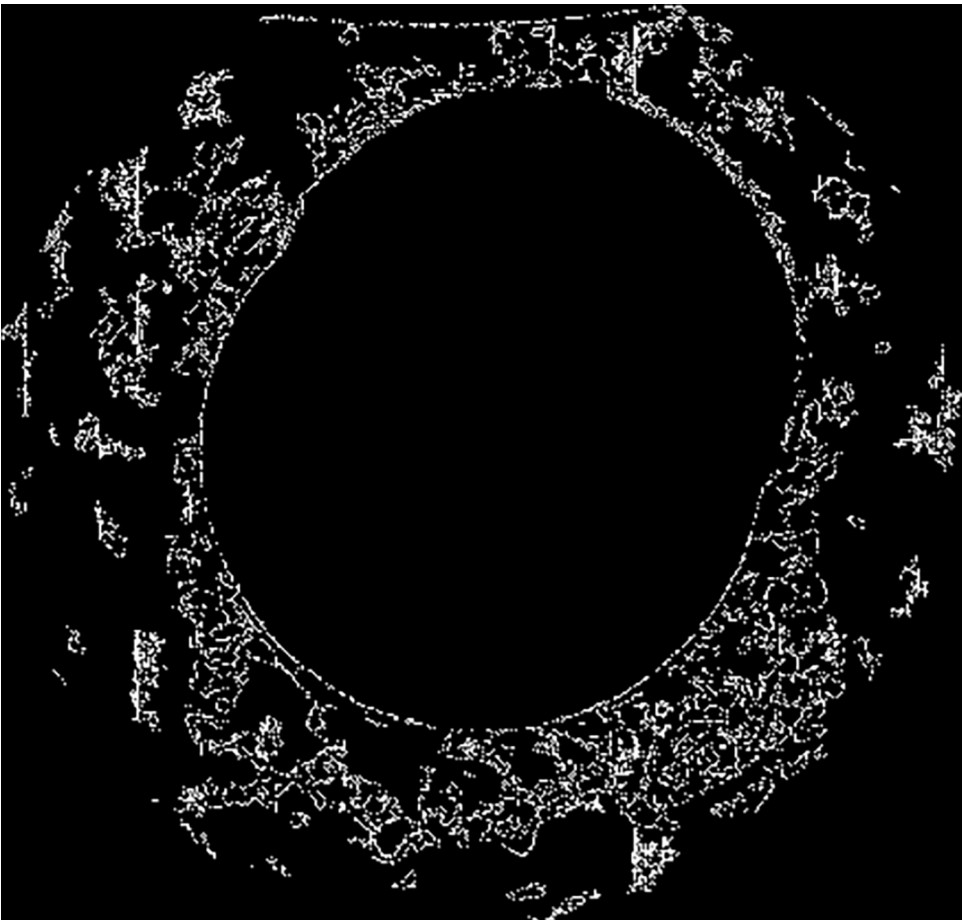

**Fig 5. Edge extraction.**

measurement, so it is necessary to filter the image to suppress the noise in the image and improve the quality of the image. Common filters include mean filter, Gaussian filter, median filter, and bilateral filter. To analyze the denoising effect of the above four image filters and their influence on the contour edge information of the ring, noise is added to the ring image, as shown in Fig 6(B). The above four filtering methods are used to filter the image with noise. Mean filtering and Gaussian filtering can remove the noise in the image well, but the edge information of the ring is blurred, which is not conducive to the edge processing in the later stage. The median filter directly blurred the whole image, and the noise was not well removed. Bilateral filtering has the best effect, which can retain the sharpness of the ring edge while removing noise. Bilateral filtering is chosen as the final image filtering method.

To expand the difference between the features of the ring and other objects, the edge information of the ring is highlighted, and the image is enhanced by Gamma transform to improve the contrast of the ring rolled part, as shown in Fig 7(A). After image enhancement, the contrast of the ring image has been significantly improved, but it contains different gray levels. To make the ring object more prominent, enhance the effective information of the ring edge, and improve the efficiency of subsequent processing, it is necessary to convert the gray figure of the ring into a binary graph. Threshold segmentation methods include global mean segmentation, Otsu threshold segmentation and iterative threshold segmentation. Due to the presence of interfering objects, the global mean segmentation method makes the interfering objects be

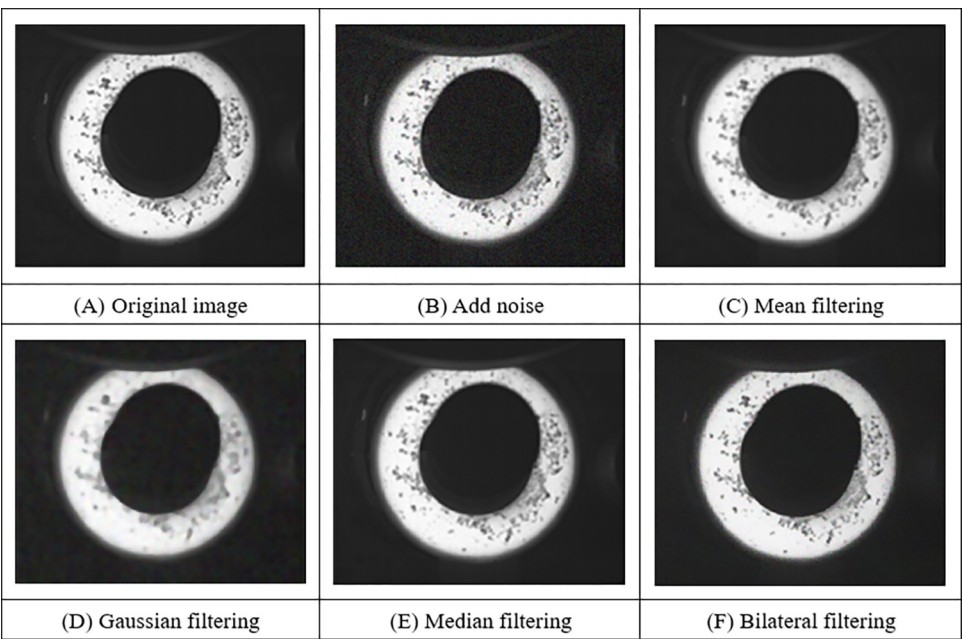

**Fig 6. Filter processing.**

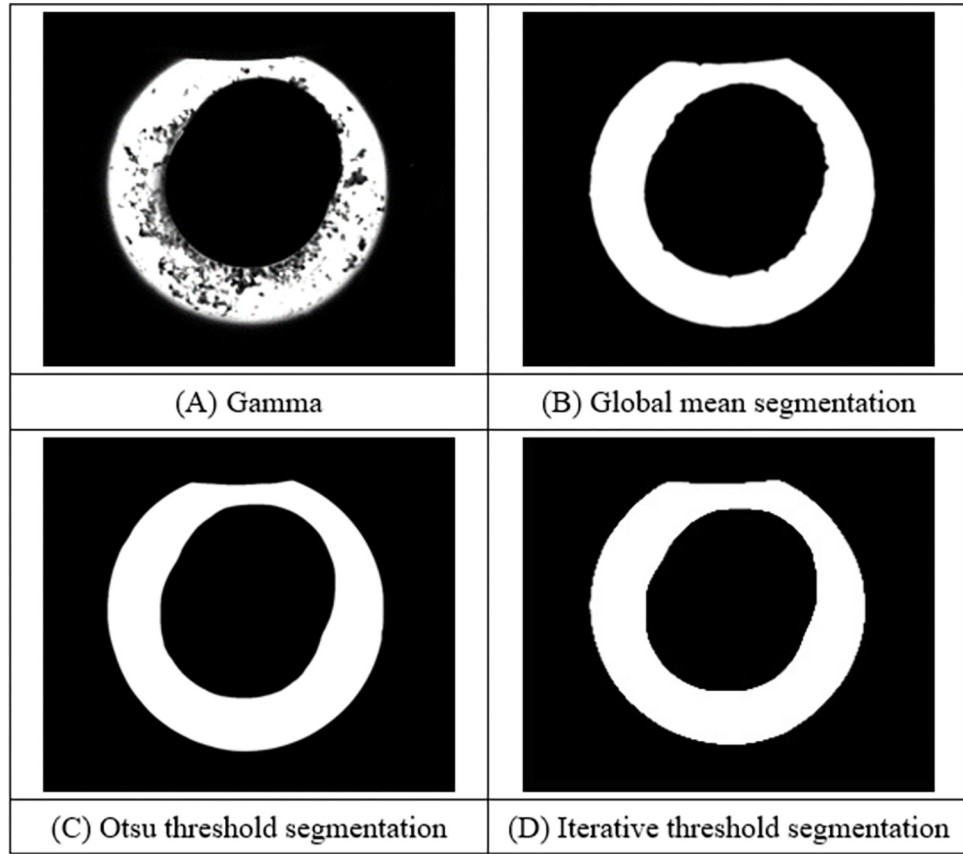

**Fig 7. Threshold segmentation.**

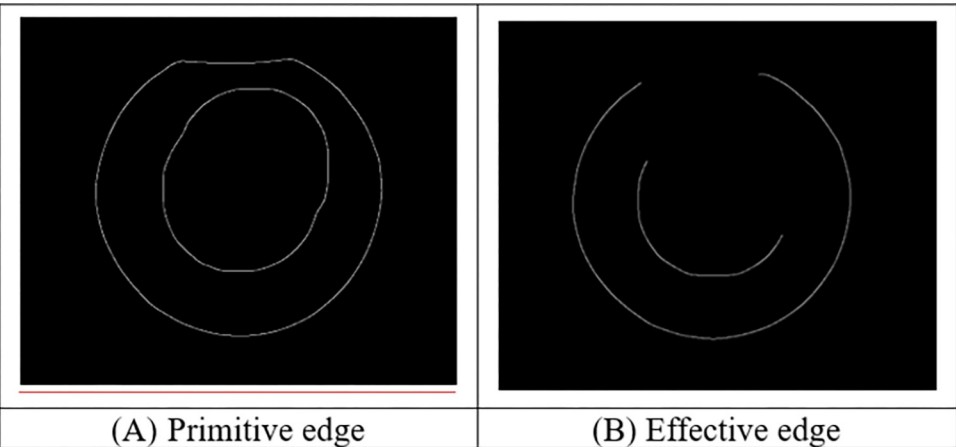

**Fig 8. Edge detection.**

divided into target objects, confusing the edge contour of the ring, which is not conducive to the subsequent processing. The iterative threshold segmentation method and the Otsu threshold segmentation method can retain the contour information of the ring better, but the iterative threshold segmentation method consumes longer time and has lower efficiency, while the Otsu threshold segmentation method has faster processing speed and better segmentation effect.

Through the above processing, the outline of the ring can be clearly obtained, as shown in Fig 8(A). However, the driving roller and the core roller in the contour are in the bonding state with the ring rolling piece, and the abnormal point will appear in the direct edge detection. To obtain the effective edge of the ring, the interference edge between the drive roll and the core roll is eliminated, as shown in Fig 8(B).

### 3.2 Traditional Zernike torque sub-pixel edge detection

For the parameter measurement of ring rolled parts, the core algorithm is two parts: edge detection and edge fitting. Edge pixel data can be gained by edge detection of ring rolling parts and edge pixel data can be fitted to output a complete ring rolling edge. Their ring rolling edges can greatly affect the final ring rolling parameter measurements.

Subpixel edge detection methods can be divided into fitting method, interpolation method and moment method according to different calculation principles. The fitting method can reduce the influence of interference points on the whole result, has a good effect on image noise suppression, and has a high localization accuracy for edge information, but the model is more complex and the solution speed is slow. The accuracy of edge information detection by interpolation method is high, and the calculation method is simple, but it is easy to be affected by noise. The moment method is less computational, more efficient in sub-pixel edge location, and less sensitive to noise, which can suppress noise interference on sub-pixel edge. Zernike moment method subpixel edge detection selects a small window near the edge contour detected at the pixel level edge, and performs Zernike polynomial fitting on the pixel intensity distribution in the window. Zernike polynomial is used to approximate the actual value of pixels in the window. Zernike polynomial is orthogonal on the unit circle. With good rotation and scaling invariance, it can obtain the local structural features of the image well and obtain more accurate edge positioning. Compared with other methods, Zernike moment method has good stability for different types of edges and can effectively deal with the interference of image noise, which is suitable for the ring visual measurement system.

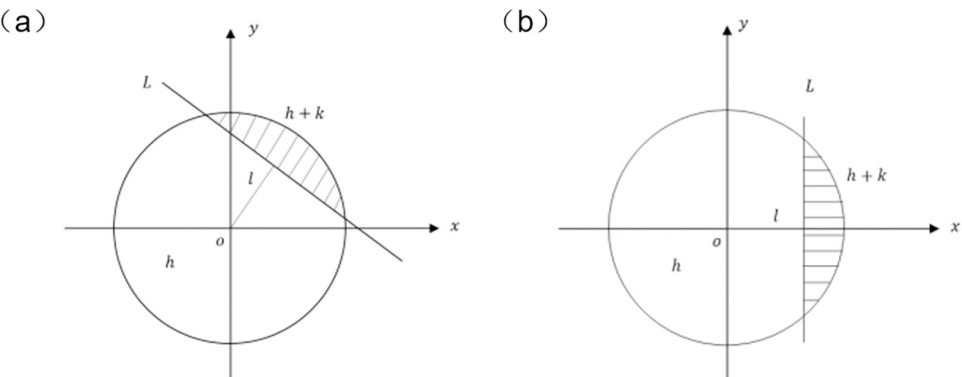

**Fig 9.** Ideal edge step edge model: (a) Initial image edge; (b) Reversed image edge.

The key of Zernike moment to realize sub-pixel positioning is to calculate four edge parameters, including rotation Angle, distance from center to edge, background gray level and step gray level. Comparison of the solved parameters and threshold settings results in accurate sub-pixel edge positioning. As shown in Fig 9, an ideal step sub-pixel edge model is set up, which includes the following parameters:

L—ideal edge of an image;

l–distance between the center and the ideal edge;

h—background grey scale; and

k—step grey scale

In the unit circle of the polar coordinate system, the n-order $m^{th}$ power orthorhombic Zernike polynomial is expressed as $V_{nm}(\rho, \theta) = R_{nm}e^{im\theta}$, where $n$ $(n\geq0)$ and $m$ are integers, $n-|m|\geq0$ is an even number, $i$ is the imaginary unit and $R_{nm}$ is a real-valued polynomial.

$$R_{nm} = \sum_{s=0}^{\frac{n-|m|}{2}} \frac{(-1)^s(n-s)!\rho^{n-2s}}{s!\left(\frac{n+|m|}{2}-s\right)!\left(\frac{n-|m|}{2}-s\right)!} \tag{1}$$

The n-order $m^{th}$ power Equation of Continuous Image $f(x, y)$ can be expressed as:

$$Z_{nm} = \frac{n+1}{\pi} \iint\limits_{x^2+y^2} f(x,y)V_{nm}^*(\rho,\theta)dx\,dy \tag{2}$$

where, $V_{nm}^*(\rho, \theta)$ is complex conjugate of $V_{nm}(\rho, \theta)$.

According to the unchanged rotation characteristics of the Zernike torque relationship exists image rotation before and after the image rotation is: $Z'_{nm} = Z_{nm}e^{-im\theta}$, and $Z_{00}$, $Z_{11}$ and $Z_{20}$ can be determined by means of three Zernike torques, whose corresponding plurals are $V_{00} = 1$, $V_{11} = x+iy$, and $V_{20} = 2x^2+2y^2-1$, respectively. The corresponding relationship can be obtained:

$$\begin{cases} Z'_{00} = Z_{00} \\ Z'_{11} = Z_{11}\,e^{i\theta} \\ Z'_{20} = Z_{20} \end{cases} \tag{3}$$

After the rotation, Zernike torques are expressed as follows:

$$Z'_{00} = h\pi + \frac{k\pi}{2} - k\sin^{-1}(l) - kl(1 - l^2)^{\frac{1}{2}} \tag{4}$$

$$Z'_{11} = \frac{2k(1 - l^2)^{\frac{3}{2}}}{3} \tag{5}$$

$$Z'_{20} = \frac{2kl(1 - l^2)^{\frac{3}{2}}}{3} \tag{6}$$

Integration of the above three equations is applied to solve out the edge parameters:

$$\begin{cases} \theta = \arctan\dfrac{Im(Z_{11})}{Re(Z_{11})} \\[2mm] l = \dfrac{Z_{00}}{Z'_{11}} \\[2mm] h = \dfrac{Z_{00} - \dfrac{k\pi}{2} + k\sin^{-1}(l) + kl(1 - l^2)^{\frac{1}{2}}}{\pi} \\[2mm] k = \dfrac{3Z'_{11}}{2(1 - l^2)^{\frac{1}{2}}} \end{cases} \tag{7}$$

While edge parameters are obtained, sub-pixel edge positioning in an ideal state is as follows:

$$\begin{bmatrix} x_s \\ y_s \end{bmatrix} = \begin{bmatrix} x \\ y \end{bmatrix} + l \begin{bmatrix} \cos\theta \\ \sin\theta \end{bmatrix} \tag{8}$$

In the actual situation, various Zernike torques are obtained by means of template and image convolution, during which expansion effect occurs. Generally, one is enlarged to two times for a template (size: N×N). Thus, the actual sub-pixel edge positioning equation is as follows:

$$\begin{bmatrix} x_s \\ y_s \end{bmatrix} = \begin{bmatrix} x \\ y \end{bmatrix} + \frac{N}{2} l \begin{bmatrix} \cos\theta \\ \sin\theta \end{bmatrix} \tag{9}$$

### 3.3 Improved Zernike torque sub-pixel edge detection

While edges are searched in a traditional Zernike torque sub-pixel edge detection, Zernike torques are necessarily calculated out for all pixels in an image because their grey scales are unknown. Their edge data are then solved based on the Zernike torque rotation invariance so that the computing burden can be large but also the calculation efficiency shall be low. The measurement real -time performances are not ideal. In case of determining whether a pixel point is at the edge, its step grey scale (k) be artificially set. If k is small, some pseudo edge points will be detected to affect the measurement accuracy of the ring rolling parts. Conversely, if k is large, some edge details may be missed to result in loss of edge pixel data of the ring rolling parts. Thus, repeated experiments are necessary for selection of the favorable step grey scale. Also, it cannot be guaranteed that the step grey scale shall be optimal.

As for our improved Zernike torque sub-pixel edge detection, coarse positioning of ring rolling edges is first carried out by means of the Canny operator and their sub-pixel edges are

then extracted to decrease calculation of unnecessary pixels but improve the edge detection efficiency.

The edge information is extracted by means of four steps (image filtering, assignment of the calculation gradient, non-maximum value inhibitory and the double threshold method) as for our Canny operator. Convolution processing of images is performed by means of the two-dimensional Gauss function $G(x, y) = \frac{1}{2\pi\sigma^2} e^{-\frac{x^2+y^2}{2\sigma^2}}$ to output a smoother image $f(x, y)$, whose horizontal and vertical partial derivatives ($f_x(x, y)$ and $f_y(x, y)$) are solved out, respectively. The gradient amplitude and direction are calculated based on application of the following equation:

$$\begin{cases} M(x, y) = \sqrt{f_x^2(x, y) + f_y^2(x, y)} \\ \theta(x, y) = \arctan\left(\frac{f_y(x, y)}{f_x(x, y)}\right) \end{cases} \tag{10}$$

Comparison of the solved gradient amplitude and the gradient of the neighboring pixels of the current point is performed to determine whether the current point is a local maximum. It is regarded as an edge point in case of a local maximum; otherwise, the gradient of the current point shall be set as zero.

For selection of the step grey scale, the OTSU method is applied to calculate the background and the maximum between-cluster variance based on the maximum step grey scale for ring rolling parts. So, the optimal step grey scale can be solved out. Compared to traditional Zernike torque sub-pixel edge detection, the edge detection time can be saved and the solved step grey scale will be more accurate.

### 3.4 Hough transformation

Under the 2D Descartes coordinates, the analysis equation of a circle is expressed as $(x - a)^2 + (y - b)^2 = r^2$, where $(a, b)$ is coordinates of the center and r is the radius. Determination of a circle requires three parameters $(a, b, \text{and } r)$. An accumulator array is required in the 3D Hough space to express all possible circles. As for each edge point $(x, y)$ in the image, all possible center positions $(a, b)$ will be traversed. As for each possible center position, calculation of the distance from the current edge pixel is regarded as a possible radius $r$, one is added at the corresponding position in the accumulator array. In the Hough space, the high-value of the accumulator corresponds to a circle in the image. A threshold is set for the accumulator. Only when the value exceeds the threshold, it is regarded as existence of a circle corresponding to the parameter in the image.

While Point P $(x_p, y_p)$ is assumed to conform to $(x_p - a)^2 + (y_p - b)^2 = r^2$, there are infinite circles through Point P, the 2D Point P is mapped into a 3D Hough space to form a 3D cone surface. All circles through Point P are mapped into the 3D Hough space to gain a 3D cone surface group, which includes a common intersection point $(a_0, b_0, c_0)$, as shown in Fig 10. After all edge points are mapped, the corresponding coordinates of the maximum in the accumulator are the center in the 2D Cartesian coordinates and the plane including the maximum in the accumulator is regarded as the radius in the 2D Cartesian coordinates.

### 3.4 RG-Hough transformation

As for the traditional Hough transformation, traversing of all edge points in the image will consume a lot of time; moreover, a false circle may occur during this process so that there shall be a certain invalid sample. There is a definite range of changes in the radius of the annulus

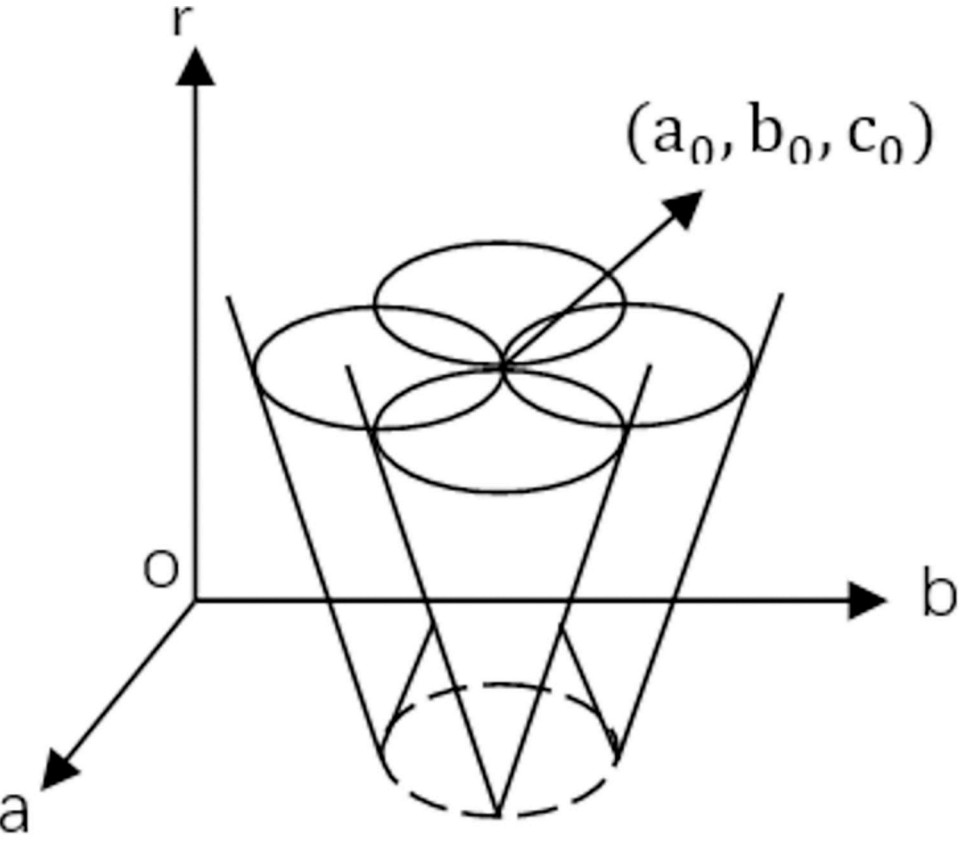

**Fig 10. 3D parameter space.**

during the rolling process, and the RG-Hough transform is proposed by combining the Hough transform and the transformation characteristics of the annulus. First, the transformation range $((R_{min}, R_{max}))$ of the ring rolling part radius is determined. While it is assumed that b may be any y-value, a can be solved out by means of the circular analytical equation $(a = y - \sqrt{r^2 - (x - b)^2})$ to acquire the Hough priority sequence value, in combination of which the radius corresponding to the $(R_{min}, R_{max})$ can be calculated out. Correspondence of the above radius and all a and b values results in determination of the center.

## 4 Experimental results and analysis

### 4.1 Experimental platform construction

For verification of the effectiveness and real time of our measurement method, a visual measurement platform was set up light of our mechanical design scheme, as shown in Fig 11. The overall size of the platform is 1500*1500*1000mm, built by profiles, divided into upper and lower parts of the space, the upper space uses the camera to real-time monitoring and measurement feedback of rolled parts, the lower space can be placed controller, rotating platform and other devices, hardware equipment includes: Industrial camera, lens, data acquisition card, control machine, light source, camera fixture, camera support frame, aluminum plate, controller, motor, heavy duty casters, rotating platform, electric telescopic rod and standard ring, combined with the actual measurement needs of the overall test platform.

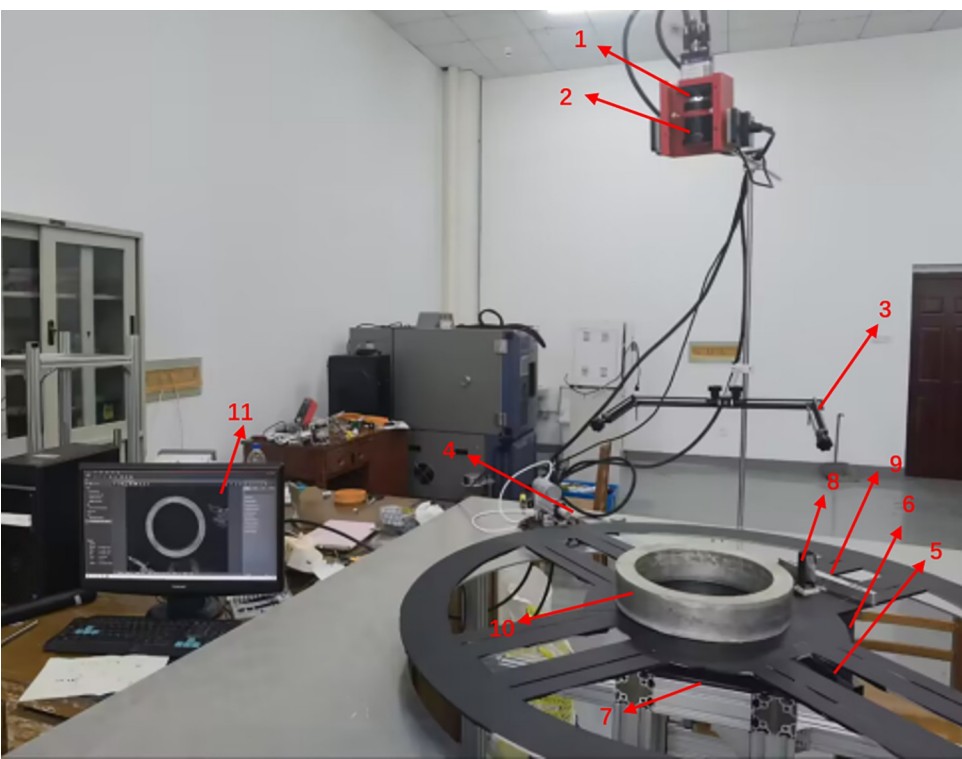

**Fig 11. Experimental platform.** 1 - Industrial camera; 2—Lens; 3—Light sources; 4—Collision block; 5—Linear motors; 6—Motorized expansion link; 7—Rotating platform; 8—Rubbing block; 9—Locate the block; 10—Ring part; and 11 –Computer.

The platform is made up of a computer (Windows10, processor: Intel(R) Core (TM) i7-11800H, CPU: 2.30GHz, memory: 16 GB), an industrial camera (model: MV-CH050) and a lens (model: MVL-KF1228) and a data acquisition card that are connected by means of a Camera Link cable (model: MV-SDR-MDR). The data acquisition card is equipped inside the controller host and real-time data acquisition of ring parts is performed by means of our visual measurement software platform. The frame rate of the camera is 140fps, and 140 images can be measured per second. The camera is fixed on the camera support frame by the camera fixture, which is placed directly above the ring rolling piece. The camera support frame can rotate 360˚, and the height and length can be adjusted to increase the measurement range of the vision system. The aluminum plate is machined to form a working platform to carry the construction of other equipment, while the machined aluminum plate is placed on the rotating platform to carry the placing of the ring, and relates to the rotating platform through the connecting part, which can drive the ring to move together. The controller can control the speed and direction of the rotating platform, and then drive the motion state of the ring. The rotating speed is 0–200 rpm, which can simulate the rotating speed of the ring under different rolling conditions. The rotating platform needs to meet the rotating speed can be continuously controlled, the maximum speed is to rotate at the speed of 1 second per revolution, and the limit load can reach 30KG under the requirement of 1 second per revolution. The rotating platform of model MTG200 is selected, and the speed is allowed to be 200rpm. The rotating platform is connected to the bearing platform of the ring through bolts. The model of the equipment connected to the motor through the motor connector to build the vision hardware platform is shown in Table 1.

**Table 1. Equipment model.**

| Name | Remarks |
|---|---|
| Industrial camera | 5 million pixels, 140fps, monochrome camera, Camera Link interface |
| Lens | Focal length: 12mm, field angle: D: 66.7˚, H:57˚, V:45˚ |
| Data acquisition card | 1720 MB/s (peak transmission bandwidth), 1600 MB/s (continuous transmission bandwidth) |
| Fixture | Adjustment accuracy: 1mm, maximum width: 100mm |
| Motor | 4.0A, 12N·M |
| Rotation platform | Speed: 0~200rpm, rotation plane deviation: ±0.005 |
| Electrical expansion link | Stroke: 100mm, 800mm, speed range: 0~180mm/s, potentiometer (optional), thrust: $\leq$ 1000kg |

To improve the intuitive experience of user operation and reduce the difficulty of user operation, the measurement software platform integrating real-time acquisition, visual measurement and result display was developed. The measurement software platform is based on the Windows operating system, using Visual Studio2019 and MFC for the design of the platform interface, and C++ language combined with OpenCV for the design of measurement algorithms. Users can call the data acquisition card and industrial camera to carry out real-time acquisition of the moving ring pieces through the visual measurement software platform, combined with the ring piece image preprocessing, edge detection and edge fitting methods, to get the ring size measurement data, and real-time display and data visualization, running the interface of the software is shown in Fig 12.

## 4.2 Camera calibration

Camera calibration is an important form of calibration procedure to ensure the accuracy and reliability of the measurement system. In this paper, Zhengyou Zhang camera calibration method is used for camera calibration experiment. Zhengyou Zhang camera calibration method uses a simple black and white high-precision checkerboard for calibration. The camera is used to collect images of the calibration board at different positions, and the checkerboard corners are extracted according to the corner coordinates in the world coordinate

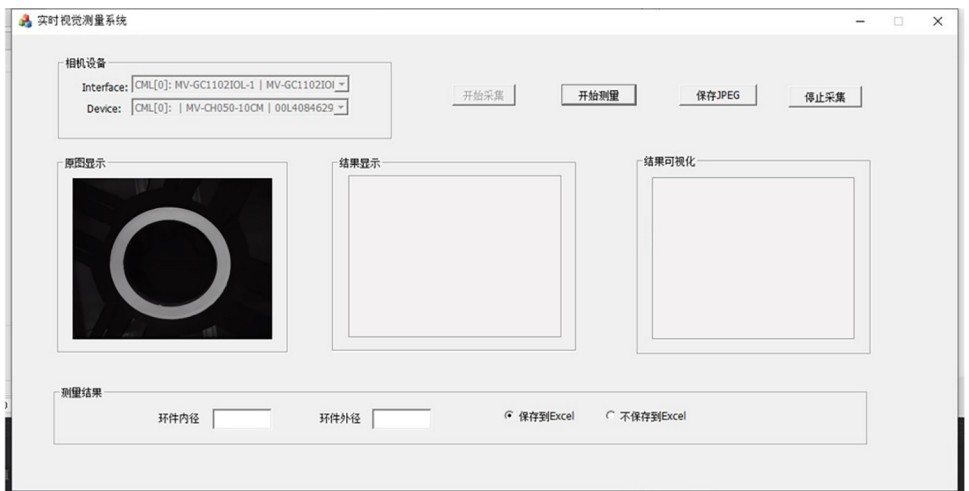

**Fig 12. Software interface.**

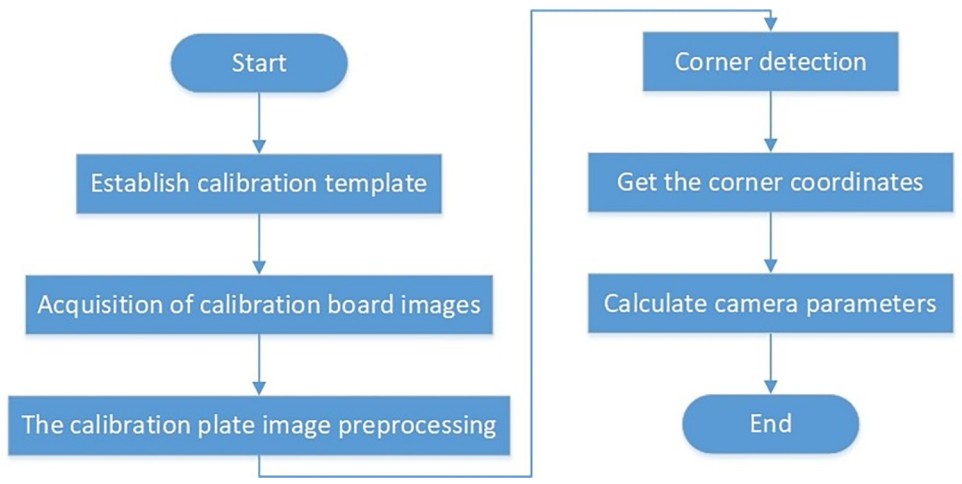

**Fig 13. Camera calibration.**

system and the corner coordinates in the image coordinate system. Find the parameters of the camera. The specific process is shown in Fig 13.

Make 28mm square checkerboard with nine rows and five columns, establish a simple calibration template, fix the camera on the visual measurement platform, adjust the camera parameters so that the camera can clearly capture the checkerboard image. In the process of camera calibration, to ensure higher calibration accuracy, it is necessary to change the Angle and position of the checkerboard, collect several images of different angles and different positions of the calibration board, and retain the images with better checkerboard corner features. By preprocessing the acquired checkerboard image, the effect of noise on the image is reduced, the effective features of the checkerboard image are highlighted, and more accurate camera parameters are obtained. Then the processed checkerboard images are detected by corner points. The coordinates of the black and white corner points in the checkerboard image are extracted, and the parameters of the camera are solved by the known coordinates of the corner points on the checkerboard in the world coordinate system and the corresponding pixel coordinate system, as shown in Table 2.

### 4.3 Experimental results and analysis

**4.3.1 Comparison experiment.** To verify the effectiveness of the improved Zernike Moment Subpixel Edge Detection, an edge detection-based comparison experiment is conducted on the same binary image using traditional Zernike Moment Subpixel Edge Detection and improved Zernike Moment Subpixel Edge Detection and some of the edge detection result maps are taken for comparison as shown in Fig 14.

**Table 2. Camera calibration result.**

| Camera parameter | Calculation result |
| --- | --- |
| $f_x$ | 1263.9632 |
| $f_y$ | 1263.7625 |
| $u_0$ | 914.9276 |
| $v_0$ | 664.0421 |
| $k_1$ | 0.0822 |
| $k_2$ | -0.0634 |

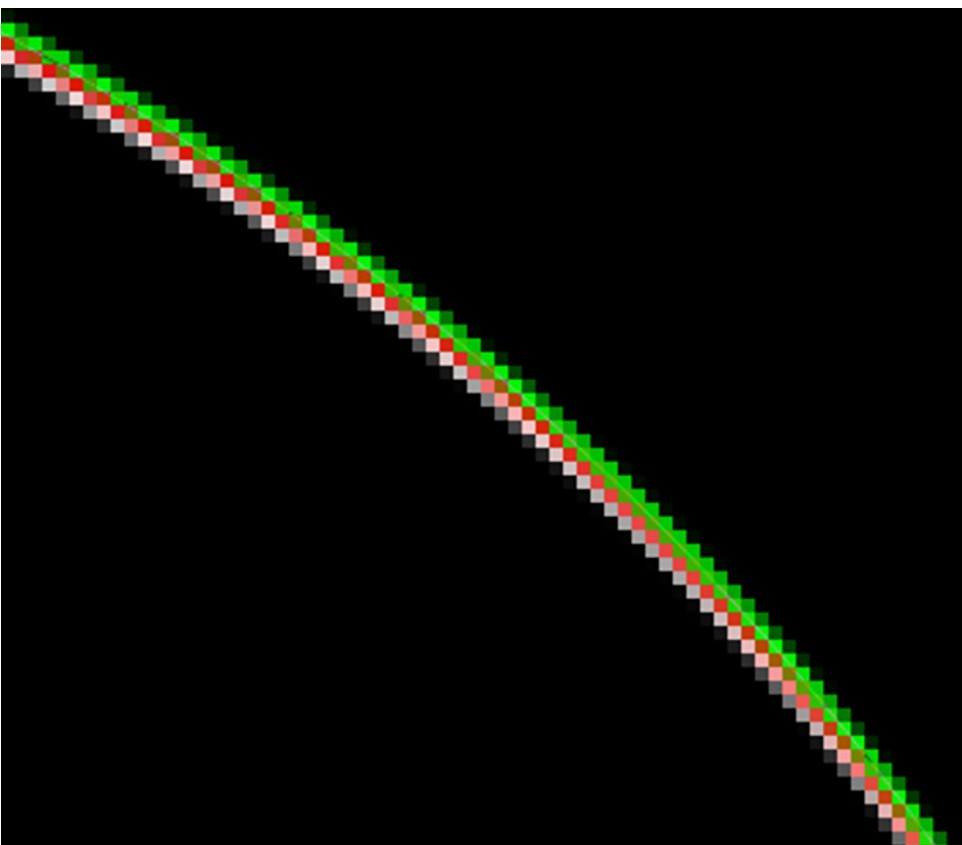

**Fig 14. Subpixel edge detection.**

In the figure, the white lines are the real edges, the green lines are the traditional Zernike moment subpixel edge detection results, and the red lines are the improved Zernike moment subpixel edge detection results. The improved Zernike moment subpixel edge detection is closer to the real value compared to the traditional Zernike moment subpixel edge detection. Five points are randomly selected and the detected coordinates are compared and analyzed and the results are shown in Table 3. The error values are visualized as shown in Fig 15.

In the figure, improved is the improved Zernike moment subpixel edge detection error value and Traditional is the traditional Zernike moment subpixel edge detection error value. The improved Zernike moment subpixel edge detection has a smaller detection error than the traditional Zernike moment subpixel edge detection and a higher accuracy of localization.

Edge fitting was performed on the obtained edge pixel data using Hough transform and RG-Hough transform. Since the circular edges in the binary image are more obvious, the edge

**Table 3. Hardware model selection.**

| Real coordinates | Traditional algorithm | Improved algorithm |
|---|---|---|
| 108, 231 | 108.45, 230.53 | 108.13, 230.75 |
| 176, 147 | 176.56, 147.78 | 176.27, 147.48 |
| 389, 76 | 388.37, 76.48 | 388.64, 76.14 |
| 547, 131 | 547.46, 131.45 | 547.27, 131.25 |
| 663, 426 | 662.38, 425.46 | 662.87, 425.85 |

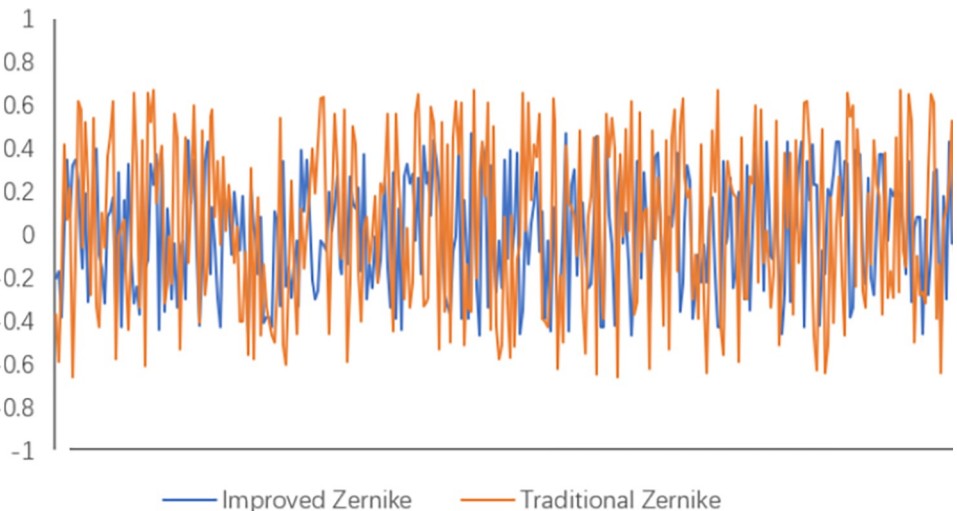

**Fig 15. Error value.**

fitting effect of the two algorithms is not much different. The time consumption of the two algorithms was counted, as shown in Fig 16, the average time consumption of the Hough transform is 152ms and the average time consumption of the RG-Hough transform is 104ms, the RG-Hough transform consumes a shorter time and has a higher detection efficiency.

**4.3.2 Measurements.** The experimental platform can measure the ring rolled parts in the range of 100mm to 1000mm, and the measurement experiment needs to consume a lot of ring wool parts, the cost is high and will cause industrial waste. Therefore, a ring rolled part with a moderate size was selected for the measurement experiment. The outer diameter of the ring rolled part was 462.12mm and the inner diameter was 315.53mm. In the process of hot rolling, there will be oxide skin in the ring rolled parts. To make the experimental scene closer to the real rolling scene, the ring parts were reformed and some burrs and dents were added to the smooth surface of the ring parts, as shown in Fig 17.

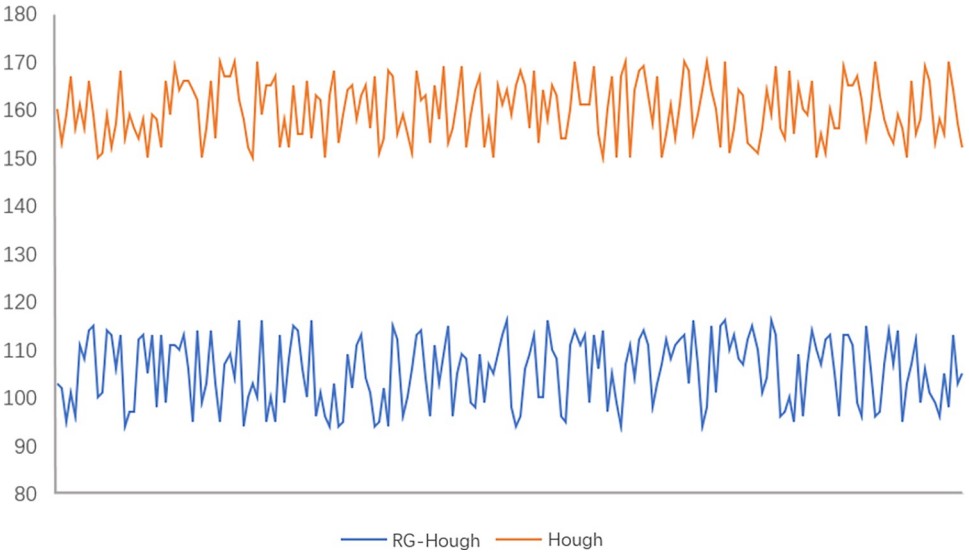

**Fig 16. Edge fitting time consuming.**

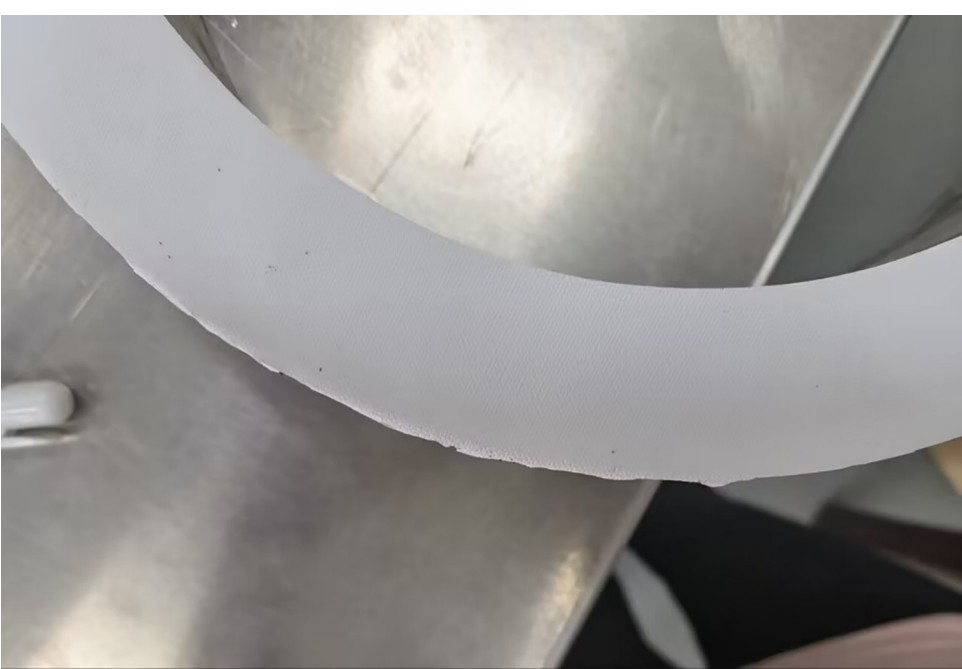

**Fig 17. Ring rolling.**

The ring is placed on the experimental platform, and the connected data acquisition card and the industrial camera are opened using the software platform. The experimental platform does not attach movement to the environment, but keeps the ring in a static state, and measures the size of the ring currently. The measurement results are shown in Table 4.

The table shows the measured values of inside and outside diameters of 10 frames and the error values of inside and outside diameters. Compared with the real values of the measured results of 10 frames, the maximum error of the inside diameter is 0.24mm and the maximum error of the outside diameter is 0.19mm, and the measured error is within the allowable range. The inner and outer diameter measurements of 400 frames were selected for visualization, as shown in Fig 18. It can be seen from the figure that the inner and outer diameter measurements of the ring are floating up and down around the true value, and the error range of the inner diameter is within ±0.25mm and the error range of the outer diameter is within

**Table 4. Static state measurement.**

| Frame | ID measurement/mm | ID error /mm | OD measurement /mm | OD error /mm |
|-------|-------------------|--------------|--------------------|--------------| 
| 1 | 315.42 | -0.11 | 461.93 | -0.19 |
| 2 | 315.42 | -0.11 | 462 | -0.12 |
| 3 | 315.63 | 0.1 | 462.28 | 0.16 |
| 4 | 315.48 | -0.05 | 462.11 | -0.01 |
| 5 | 315.75 | 0.22 | 462.25 | 0.13 |
| 6 | 315.6 | 0.07 | 462.2 | 0.08 |
| 7 | 315.67 | 0.14 | 462.09 | -0.03 |
| 8 | 315.37 | -0.16 | 462.15 | 0.03 |
| 9 | 315.77 | 0.24 | 462.2 | 0.08 |
| 10 | 315.32 | -0.21 | 462.07 | -0.05 |

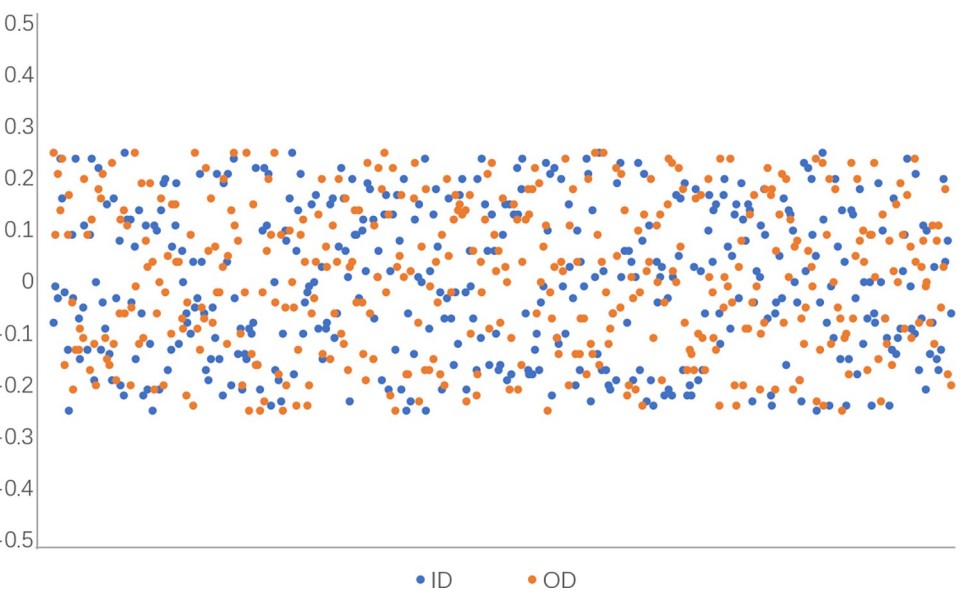

**Fig 18. Measurement error.**

±0.25mm, which verifies the effectiveness of the visual measurement system for the measurement of the ring.

This platform is aimed at the parameter measurement of ring rolled parts in the rolling process. To verify the effectiveness of this vision system for the measurement of high-speed ring rolled parts, the experimental platform is used to adjust the state of the ring rolled parts, so that the ring rolled parts rotate at 5 rpm, 20 rpm and 40 rpm, and the impact block and top block are used to interfere with the ring rolled parts. The visual measurement results of ring rolled parts at different speeds were measured through the visual measurement system, and 10 frames of ring rolled parts were randomly selected for display, as shown in Tables 5–7.

Tables 5–7 shows the od and id measurements and od and id errors in frame ten. It can be seen from the measurement results of the inner and outer diameter of the table that the increased motion state has little influence on the measurement of the ring, the overall error value does not increase greatly, and the error size of different speeds does not fluctuate greatly. 400 frames of measurement results were taken for visualization, as shown in Fig 19. It can be

**Table 5. Measurements @5rpm.**

| Frame | ID measurement/mm | ID error /mm | OD measurement /mm | OD error /mm |
|-------|-------------------|--------------|--------------------|--------------| 
| 1 | 315.47 | -0.06 | 462.08 | -0.04 |
| 2 | 315.43 | -0.1 | 462.03 | -0.09 |
| 3 | 315.39 | -0.14 | 461.97 | -0.15 |
| 4 | 315.29 | -0.24 | 462.02 | -0.1 |
| 5 | 315.35 | -0.18 | 461.99 | -0.13 |
| 6 | 315.37 | -0.16 | 462.22 | 0.1 |
| 7 | 315.48 | -0.05 | 462.26 | 0.14 |
| 8 | 315.7 | 0.17 | 462.23 | 0.11 |
| 9 | 315.61 | 0.08 | 462.2 | 0.08 |
| 10 | 315.45 | -0.08 | 462.13 | 0.01 |

**Table 6. Measurements @20rpm.**

| Frame | ID measurement/mm | ID error /mm | OD measurement /mm | OD error /mm |
|---|---|---|---|---|
| 1 | 315.58 | 0.05 | 462.06 | -0.06 |
| 2 | 315.64 | 0.11 | 462.11 | -0.01 |
| 3 | 315.61 | 0.08 | 462.3 | 0.18 |
| 4 | 315.5 | -0.03 | 462.21 | 0.09 |
| 5 | 315.53 | 0 | 462.12 | 0 |
| 6 | 315.44 | -0.09 | 462.09 | -0.03 |
| 7 | 315.51 | -0.02 | 462.08 | -0.04 |
| 8 | 315.48 | -0.05 | 462.24 | 0.12 |
| 9 | 315.4 | -0.13 | 462.1 | -0.02 |
| 10 | 315.59 | 0.06 | 462.03 | -0.09 |

**Table 7. Measurements @40rpm.**

| Frame | ID measurement/mm | ID error /mm | OD measurement /mm | OD error /mm |
|---|---|---|---|---|
| 1 | 315.59 | 0.06 | 462.02 | -0.1 |
| 2 | 315.5 | -0.03 | 462.08 | -0.04 |
| 3 | 315.61 | 0.08 | 462.12 | 0 |
| 4 | 315.56 | 0.03 | 462.14 | 0.02 |
| 5 | 315.52 | -0.01 | 462.22 | 0.1 |
| 6 | 315.47 | -0.06 | 462.13 | 0.01 |
| 7 | 315.51 | -0.02 | 462.1 | -0.02 |
| 8 | 315.52 | -0.01 | 462.04 | -0.08 |
| 9 | 315.58 | 0.05 | 462.01 | -0.11 |
| 10 | 315.48 | -0.05 | 462.03 | -0.09 |

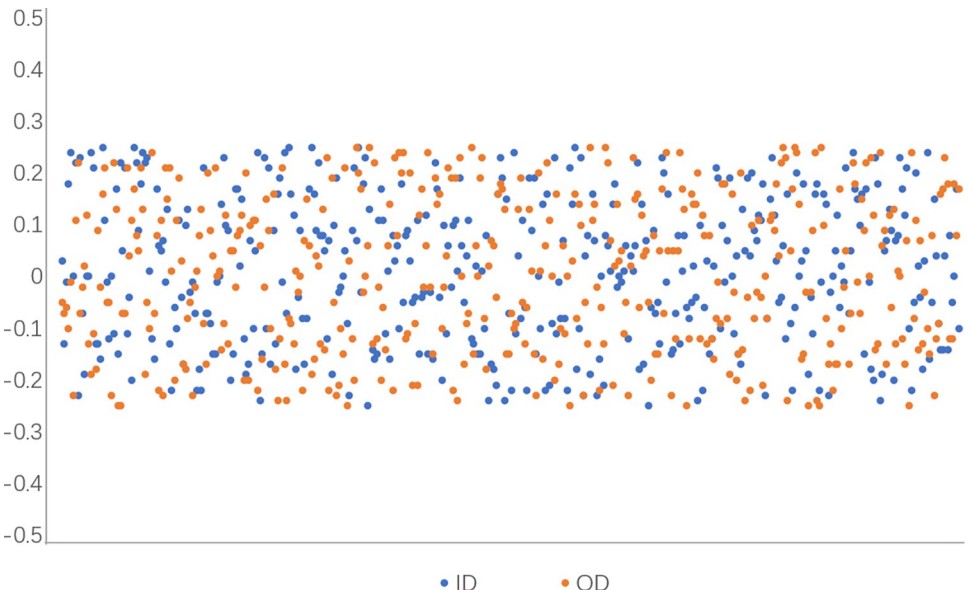

**Fig 19. Measurement error.**

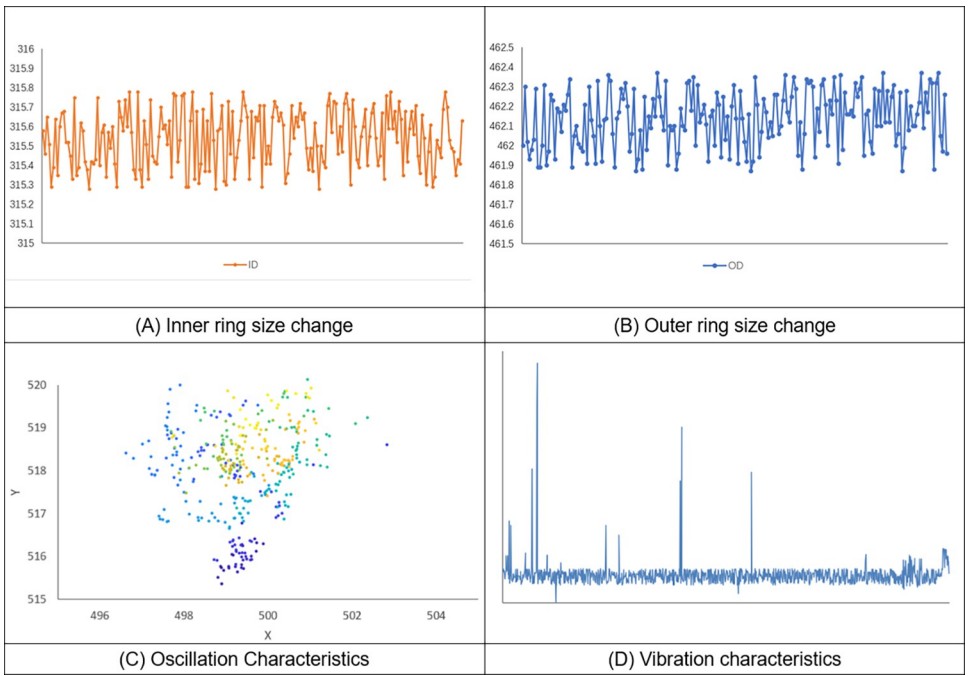

**Fig 20. Visualization of parameter measurements.**

seen from the figure that the visual measurement results are relatively stable for different speeds, which verifies the stability of the visual measurement system.

Based on the measured parameter data of the ring rolled parts, the size change, center swing characteristics and vibration characteristics of the ring rolled parts were calculated and analyzed, and visualized, as shown in Fig 20.

### 4.4 Error analysis

The above measurements show that there are still errors although our can visual measurements ensure a certain measurement accuracy. There are many factors such as physical limitations of hardware, software processing techniques and unstable environmental variables that affect the accuracy during the visual measurement process. From the analysis of the entire measurement process, errors are primarily coming from 2 sources.

(1) Camera calibration error: our visual measurement system is calibrated by means of a calibration board. Its physical dimension accuracy, uniformity and reflection or scattering of its material surface and the checkerboard corner clarity may affect not only the accuracy of the camera calibration but also the conversion between the pixel and real physical length units. For performance of a high-precision camera calibration, a high-quality and accurate calibration board is necessary.

(2) Optical lens error: the lens cannot guarantee the accuracy between the connection and fixing of each part during the manufacturing and installation process. There is a certain degree of assembly errors. Thus, the final measurement results may be affected. A camera and lens shall be selected as much as possible in the model selection stage.

(3) Environment error: From the camera calibration to the visual measurement phase, the environment lights, temperature, and humidity may change and there may be reflection and obstruction. Also, measurement operations may not be standardized. Thus, the measurement

accuracy may be affected. Those measures such as application of a stable light source and measurement environment isolation may be taken to eliminate the environment errors.

## 5 Conclusions

In view of the cases that dynamic parameter measurement of ring rolled parts cannot be performed in real time and feedback control cannot be timely during a ring part rolling process, a ring rolling part visual measurement platform is designed and an improved Zernike torque sub-pixel edge detection algorithm is presented to detect the edge data information of ring rolled parts, whose edges are fitted by means of our improved Hough transformation to gain rolling parameter measurements. Simulation results indicate that our improved Zernike torque sub-pixel edge detection has smaller errors. Comparison of periods for two Hough transformation algorithms indicates that our improved Hough transformation algorithm save time and is favorable for improvement of real-time parameter measurements of ring rolling parts. Ring rolling parts (OD: 462.12mm, ID: 315.53mm) are measure. Their errors are within ±0.25mm and the average consuming time is 104ms/frame. Their measurement errors are within the allowable ranges and the speed can meet the requirements of real -time detection. In actual industrial production, there are also parts with other shapes other than rings, and the types of experimental samples can be enriched in the future, and the corresponding algorithms can be studied to improve the versatility of the vision measurement system.

## Supporting information

**S1 File.**
(XLSX)

## Author Contributions

**Formal analysis:** Xiaoge Fu.

**Methodology:** Xiaoge Fu, Zhijiang Zuo.

**Software:** Han Li.

**Supervision:** Zhijiang Zuo.

**Validation:** Libo Pan.

**Writing – original draft:** Xiaoge Fu.

**Writing – review & editing:** Han Li, Zhijiang Zuo, Libo Pan.

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
