## [Decision Letter · Decision Letter 0]

9 Jan 2024

PONE-D-23-34868Study of Real-time Parameter Measurement of Ring Rolling Pieces Based on Machine VisionPLOS ONE

Dear Dr. Fu,

Thank you for submitting your manuscript to PLOS ONE. After careful consideration, we feel that it has merit but does not fully meet PLOS ONE’s publication criteria as it currently stands. Therefore, we invite you to submit a revised version of the manuscript that addresses the points raised during the review process.

We look forward to receiving your revised manuscript.

Kind regards,

Waqas Saleem, Ph.D

Academic Editor

PLOS ONE

Journal Requirements:

5. We note that your Data Availability Statement is currently as follows: [All relevant data are within the manuscript and its Supporting Information files.]

7. Please clarify the number of Figures uploaded in your manuscript and PDF file. 

8. Please include a copy of Table 6 which you refer to in your text on page 23.

Reviewers' comments:

Reviewer's Responses to Questions

**Comments to the Author**

1. Is the manuscript technically sound, and do the data support the conclusions?

Reviewer #1: Partly

2. Has the statistical analysis been performed appropriately and rigorously? 

Reviewer #1: No

3. Have the authors made all data underlying the findings in their manuscript fully available?

Reviewer #1: Yes

4. Is the manuscript presented in an intelligible fashion and written in standard English?

Reviewer #1: No

5. Review Comments to the Author

Reviewer #1: Reviewer comments to authors:

1. The introduction section should be enhanced by incorporating insights from prior research, and it would benefit from the inclusion of recent citations pertaining to ring measurements. You need to look for and add related citation as DOI below” https://doi.org/10.1016/j.measurement.2022.111072, https://doi.org/10.1007/s00170-022-10359-0, https://doi.org/10.1007/s11740-023-01232-4, https://doi.org/10.1371/journal.pone.0292814, https://doi.org/10.3390/app11062601, https://doi.org/10.3390/s23177443, “

2. In the methodology section:

a. Can you provide more comprehensive details about the instruments and devices utilized for ring measurements in the methodology section?

b. How would including information about the measurement equipment's accuracy and calibration procedures contribute to strengthening the overall robustness of the study?

c. Could you offer clarification on the sample size and variability in the experiments to ensure a clearer understanding of the research parameters?

d. In what ways can you elaborate on the data collection process, including the frequency of measurements and the specific conditions under which they were conducted?

3. It is necessary to verify the formatting of the references and the spacing between sentences and references. Such as “measurement[16] line 60, 74 , and Lizhou[17] line 79”.

4. Some typos need to change “information(Fig 5) line 206”.

5. It is necessary to discuss how the algorithms and pre-processing steps selected affect the overall quality of the results. How do these steps help to improve measurement accuracy or expand the system's capabilities? This link should be highlighted explicitly.

6. Consider discussing the performance of various algorithms, if applicable. Is it possible to find algorithms that outperform others in specific scenarios? Exploring these comparative aspects can provide insights into the decision-making process for algorithm selection.

7. Recommend potential future research directions, such as investigating the method's extension to address challenges associated with various types of metrology measurements or exploring how it interacts with modern manufacturing systems.

8. The manuscript needs to be proofread to be more suitable for publishing.

6. PLOS authors have the option to publish the peer review history of their article (what does this mean?). If published, this will include your full peer review and any attached files.

Reviewer #1: **Yes: **Yazid Saif

---

## [Author Response · Author response to Decision Letter 0]

25 Jan 2024

Response to Reviewers

Dear Editors and Reviewers:

Thank you for your letter and for the reviewer(s)’ comments concerning our manuscript entitled “Study of Real-time Parameter Measurement of Ring Rolling Pieces Based on Machine Vision” (Manuscript ID PONE-D-23-34868). Those comments are all valuable and helpful for revising and improving our paper. We have studied comments carefully and have made correction which we hope to meet with approval. Revised parts are marked in red in the paper. The main corrections in the paper and the responses to the reviewers are as follows: 

1. Response to comment: (The introduction section should be enhanced by incorporating insights from prior research, and it would benefit from the inclusion of recent citations pertaining to ring measurements.)

Response: Thank you for your advice. In the introduction, we add the research of Zheng Lu, Yu-cun Zhang, Bangguo Wang, Xianbin Fu, Yazid Saif and others on the measurement of ring parts, and add relevant citations.

2. In the methodology section:

a. Response to comment: (Can you provide more comprehensive details about the instruments and devices utilized for ring measurements in the methodology section?)

Response: Thank you very much. We have included a detailed description of the mechanical structure in "2.2 Mechanical structure design ", which explains the function of each module. In "4.1 Experimental platform construction", the hardware and equipment parameters required by the experiment platform are comprehensively introduced.

b. Response to comment: (How would including information about the measurement equipment's accuracy and calibration procedures contribute to strengthening the overall robustness of the study?)

Response: Thank you for your valuable advice. We have added the section "3.2 Camera calibration" to the text. Camera calibration is an important form of calibration procedure to ensure the accuracy and reliability of the measurement system. The process and result of camera calibration are also given.

c. Response to comment: (Could you offer clarification on the sample size and variability in the experiments to ensure a clearer understanding of the research parameters?)

Response: Thank you for your comments on our articles. According to your suggestion, we explained the reason for selecting the test ring in the measurement experiment, and carried out several measurements on the sample. The validity and stability of the experiment are verified by comparative analysis of the different conditions that affect the measurement results.

d. Response to comment: (In what ways can you elaborate on the data collection process, including the frequency of measurements and the specific conditions under which they were conducted?)

Response: Thank you for your advice. We have added detailed information about the vision equipment, including the measurement frequency and conditions, to the experimental platform construction.

3. Response to comment: (It is necessary to verify the formatting of the references and the spacing between sentences and references. Such as “measurement[16] line 60, 74 , and Lizhou[17] line 79”.)

Response: Thanks for your careful reading, we have verified the format of the references and cited them in the "PLoS" style using EndNote X9.

4. Response to comment: (Some typos need to change “information(Fig 5) line 206”.)

Response: Sorry, we have revised the article and corrected the part with mistakes.

5. Response to comment: (It is necessary to discuss how the algorithms and pre-processing steps selected affect the overall quality of the results. How do these steps help to improve measurement accuracy or expand the system's capabilities? This link should be highlighted explicitly.)

Response: Thank you very much for your advice. We added the result graph without image preprocessing in the "3.1 Image Preprocessing" section, and discussed the influence of no preprocessing steps on the measurement results. At the same time, literature [23] was added to further prove the importance of pre-treatment steps.

6. Response to comment: (Consider discussing the performance of various algorithms, if applicable. Is it possible to find algorithms that outperform others in specific scenarios? Exploring these comparative aspects can provide insights into the decision-making process for algorithm selection.)

Response: Thank you for your advice. We added the comparison of key algorithms in "3.1 Image Preprocessing", showed the comparison result graph, and expounded the basis of algorithm decision-making.

7. Response to comment: (Recommend potential future research directions, such as investigating the method's extension to address challenges associated with various types of metrology measurements or exploring how it interacts with modern manufacturing systems.)

Response: Thank you for your helpful advice. We have added a vision for the future at the end of the "5 Conclusions" section.

8. Response to comment: (The manuscript needs to be proofread to be more suitable for publishing.)

Response: Thank you for your advice. We have proofread the article and made changes to the relevant positions.

Thank you again for your positive comments and valuable suggestions to improve the quality of our manuscript.

Sincerely yours.

Wish you all the best!

Sincerely yours,

Xiaoge Fu, Han Li, Zhijiang Zuo, and Libo Pan

---

## [Editor Report · Decision Letter 1]

29 Jan 2024

Study of Real-time Parameter Measurement of Ring Rolling Pieces Based on Machine Vision

PONE-D-23-34868R1

Dear Dr. Fu,

We’re pleased to inform you that your manuscript has been judged scientifically suitable for publication and will be formally accepted for publication once it meets all outstanding technical requirements.

Kind regards,

Waqas Saleem, Ph.D

Academic Editor

PLOS ONE

---

## [Editor Report · Acceptance letter]

13 Feb 2024

PONE-D-23-34868R1 

PLOS ONE

Dear Dr. Fu, 

I'm pleased to inform you that your manuscript has been deemed suitable for publication in PLOS ONE. Congratulations! Your manuscript is now being handed over to our production team.

Kind regards, 

on behalf of

Dr. Waqas Saleem 

Academic Editor

PLOS ONE